# Interactions of Oxytocin and Dopamine—Effects on Behavior in Health and Disease

**DOI:** 10.3390/biomedicines12112440

**Published:** 2024-10-24

**Authors:** Maria Petersson, Kerstin Uvnäs-Moberg

**Affiliations:** 1Department of Endocrinology, Karolinska University Hospital, 171 76 Stockholm, Sweden; 2Department of Molecular Medicine and Surgery, Karolinska Institutet, 171 77 Stockholm, Sweden; 3Department of Applied Animal Science and Welfare, Swedish University of Agricultural Sciences, 532 31 Skara, Sweden

**Keywords:** oxytocin, dopamine, behavior, social interaction, feeding, depression, anxiety, autism spectrum disorders, schizophrenia

## Abstract

The hypothalamic neuropeptide and hormone oxytocin are of fundamental importance for maternal, social, and sexual behavior. Deviations in oxytocin levels have also been associated with anxiety, autism spectrum disorders (ASD), depression, ADHD (attention deficit hyperactivity disorder), and schizophrenia. Both oxytocin and dopamine are often considered reward- and feel-good hormones, and dopamine is associated with the above-mentioned behaviors and, and dopamine is also associated with the above-mentioned behaviors and disorders. Although being structurally totally different, oxytocin, a peptide, and dopamine, a monoamine, they have a number of similar effects. They are synthesized both in the brain and in the periphery, and they affect each other’s release and receptors. In addition, oxytocin and dopamine are released in response to, for example, social interaction, sex, feeding, and massage. This review discusses interactions between oxytocin and dopamine with a specific focus on behavioral effects and possible roles of oxytocin and dopamine in various mental disorders and functional diversities.

## 1. The History and Discovery of Oxytocin and Dopamine

In 1894, Oliver and Schäfer were the first to discover that extracts from the adrenal gland induced a rise in blood pressure [1]. The effect was found to be mediated by adrenaline, the endpoint in the synthesis of the catecholamine hormones. One year later, Oliver and Schäfer discovered that vasoconstriction could also be induced by extract from the pituitary [2]. However, this was not an effect by adrenaline but by vasopressin (structurally related to oxytocin and released from the neurohypophysis as well), and in 1906, Dale reported that extracts from the pituitary gland, in addition, induced uterine contraction [3]. This effect was mediated by the hormone oxytocin, which was also found to stimulate the milk ejection reflex during breastfeeding [4].

At that time, it was thought that dopamine was only an inactive intermediate in the catecholamine synthesis towards noradrenaline and adrenaline [5]. But with the discovery of dopamine in the brain in 1957–1959, this opinion changed [6,7,8].

In 1953, du Vigneaud sequenced and synthesized oxytocin and also vasopressin, which differs from oxytocin by only two amino acids. These were the first peptides ever to be sequenced and synthesized, and Du Vigneaud was awarded the Nobel Prize in 1955 for this discovery [9].

About 20 years later, it was found that oxytocin, besides being a hormone, is also a neurotransmitter released from a broad spectrum of nerves emanating from both magnocellular and parvocellular neurons within the supraoptic nucleus (SON) and the paraventricular nucleus (PVN), which project to other areas of the brain, brainstem, and spinal cord (see, for example, [10,11]).

Dopamine is also related to a Nobel Prize. Arvid Carlsson was awarded the Nobel Prize in 2000 because of his discovery of dopamine in the brain and the finding that dopamine is an active neurotransmitter.

In addition to this parallel history, oxytocin and dopamine have a lot of effects and characteristics in common. Despite being totally structurally different, oxytocin is a peptide and dopamine is a monoamine; they are both released in response to social interaction, feeding, sex, and massage and are considered reward, feel-good, and love hormones. Deviations in their levels have been associated with, for example, depression, anxiety, obsessive–compulsive disorders (OCD), autism spectrum disorders (ASD), attention deficit hyperactivity disorders (ADHD), and schizophrenia.

Moreover, oxytocin and dopamine influence each other’s release. Dopamine receptors are located on the oxytocin neurons within the PVN, and dopamine has been shown to stimulate the release of oxytocin [12]. Vice versa, oxytocin releases dopamine, and when an oxytocin antagonist was administered intracerebroventricularly (ICV), the release of dopamine in response to a dopamine agonist was diminished [13]. Both oxytocin and dopamine are released in response to a number of stimuli and influence several other hormones and neurotransmitters. A lot of diverse effects are induced or modulated by oxytocin and dopamine, and they both have receptors in several areas both within and outside the brain. Both the oxytocin receptor (only one main type characterized) and the dopamine receptors are G-protein-coupled transmembrane receptors, and they are often located closely to each other in the same brain areas. Oxytocin and dopamine modulate, for example, metabolism, the cardiovascular system, renal function, and the inflammatory response, but maybe of most importance, they both modulate behavior, which is the focus of the present review. Typical characteristics of oxytocin and dopamine are summarized in Table 1.

## 2. Oxytocin and Dopamine in the Brain

### 2.1. Neurons and Pathways

Oxytocin is produced in two hypothalamic nuclei: the SON and the PVN. Oxytocin neurons project to many areas with dopaminergic neurons or areas reached by dopaminergic neurons, such as the ventral tegmental area (VTA), striatum, hippocampus, amygdala, nucleus accumbens, prefrontal cortex, olfactory bulb, brain stem, and spinal cord. The oxytocinergic neurons also have a widespread network within the hypothalamus. Primarily, the parvocellular neurons within the PVN, as well as axon collaterals from the magnocellular neurons, project to other areas within the central nervous system (CNS). In addition, oxytocin can be released from the dendrites both from the parvo- and magnocellular neurons in the PVN and SON, which makes a widespread release of oxytocin within the central nervous system possible [10,14,15,16,17].

Dopamine has several distinct pathways within the brain. The nigrostriatal pathway, which runs from the substantia nigra to the striatum, is mainly associated with movements and motor activity but also plays a role in cognition and learning. The mesolimbic/mesocortical pathways originate in the VTA and project to the amygdala, nucleus accumbens, gyrus cingulate, hippocampus, olfactory bulb, and areas within the cortex involved in, for example, motivation, social behavior, learning, and reward. The tuberoinfundibular pathway projects from neurons in the nucleus arcuatus/periventricular nucleus to the mediana eminentia, releasing dopamine into the hypophyseal portal system to inhibit prolactin release [18]. There is also a less well-known pathway named the incertohypothalamic dopamine pathway that projects to the lateral septal nuclei and other areas within the hypothalamus, including the SON and PVN, where oxytocin is produced [19]. Besides these pathways within the brain, there is also a pathway from the hypothalamus to the spinal cord (for a review, see, for example, Baskerville and Douglas [20]). Additionally, there are dopaminergic neurons locally spread, for example, within the PVN [21]).

Dopaminergic neurons influence oxytocin release, and dopamine receptors have been demonstrated in oxytocin neurons. Conversely, oxytocin has been found to increase dopamine in, for example, the medial preoptic area, the amygdala, and the VTA, where oxytocin receptors are also located (see, for example [20,22,23]).

It is important to remember that oxytocin is also a hormone and is released from the neurohypophysis into the circulation, and that only small amounts of oxytocin pass the blood–brain barrier. Often, but not always, there is a parallel release of oxytocin into the circulation and into the CNS. For example, in a study by Engelmann et al., oxytocin in response to a social defeat increased in certain areas within the brain, whereas it was unchanged in peripheral blood [24]. Nor does dopamine pass the blood–brain barrier, which means that dopamine produced within the brain is separated from the dopamine produced in the periphery (for example, the gastrointestinal tract, kidneys, and adrenal glands).

### 2.2. Interactions Between Oxytocin and Dopamine

Oxytocin and dopamine are released in response to feeding, as well as sexual and social interaction in both humans and animals. They are also released in response to various other behaviors and habits such as game playing, performing exercise, listening to music, different kinds of abuse, and food intake, but sometimes also in response to starvation [20,25,26]. Drugs like amphetamine and cocaine have direct effects on dopamine release, and some studies have reported changes also in oxytocin levels in response to these drugs as well as alcohol. Oxytocin has even been suggested as a treatment for drug addiction (for a review, see, for example, McGregor and Bowen [27]). However, these effects regarding drugs and alcohol are beyond the scope of the present review.

Both oxytocin and dopamine are related to reward and pleasure, and they are both released in response to afferent vagal stimulation, touch, massage (afferent stimulation of sensory nerves), and meditation [26,28,29]. As early as 1938, Chang et al. showed an oxytocin release in response to vagal stimulation [30]. Whether oxytocin levels in response to vagal stimulation may differ between different brain areas is, to our knowledge, not known, but when it comes to dopamine, electrophysiological experiments revealed a significant decrease in the firing of dopamine neurons within the VTA after long-term stimulation of the vagal nerve, whereas dopamine levels were increased in the prefrontal cortex and nucleus accumbens [31]. In contrast, lesions of the vagal nerve in rodents have been demonstrated to decrease post-ingestive dopamine activation in the VTA [32]. The mechanism behind the increase in oxytocin levels in response to dopamine is complex and has mainly been linked to dopaminergic 2 (D2) receptors. However, D1 and D3 receptors have also been demonstrated in the oxytocin neurons [33]. In experiments using electrophysiology, D1 and D3 receptors seem to activate the oxytocinergic neurons as well [34,35], and when different dopamine-agonists were administered to rats, the increase in oxytocin levels was observed in response to a combined D2/D3-agonist and not in response to a D2-agonist only [36].

As mentioned above, oxytocin increases dopamine levels in several brain areas, such as the medial preoptic area, the amygdala, and the VTA. These areas are all provided with oxytocin receptors.

Besides increasing each other´s release, the oxytocin receptor and dopamine receptor can interact directly. Both oxytocin and dopamine receptors have been demonstrated in, for example, the VTA, the nucleus accumbens, and the amygdala [23,37,38], and in the nucleus accumbens and the amygdala, oxytocin and D2-receptors have been shown to interact through forming heterocomplexes and thereby changes in the intracellular G-protein-coupled signaling [39,40].

There are several studies demonstrating interactions between oxytocin and dopamine also in humans. In a study with fMRI, there was an increase in activity within the VTA in response to intranasally administered oxytocin and specific cues [41]. In another study, the activity within both the VTA and the nucleus accumbens increased in response to oxytocin administered intranasally [42]. More studies will be discussed below.

## 3. Behavioral Effects of Oxytocin and Dopamine

Both oxytocin and dopamine have important effects on various types of behavior, and both oxytocin and dopamine knockout or deficient mice have disrupted social behavior, including maternal and feeding behavior. Below, we will discuss behaviors and disorders associated with social ability and social behavior and their possible relationship to oxytocin and dopamine.

### 3.1. Parental Behavior

The first behavioral effects of oxytocin that were described were the effects on maternal behavior and bonding between mother and offspring/child [43,44]. Oxytocin released during parturition, breastfeeding, and in response to the early skin-to-skin contact between mother and child is necessary for these effects, including bonding. A release of oxytocin in response to early skin-to-skin contact has also been demonstrated in fathers [45,46]. Even if oxytocin might be considered the most important substance for parental behavior, dopamine is of fundamental importance, too. In fact, oxytocin and dopamine are acting together. After birth, an increase in oxytocin receptor density occurs in many areas of the brain, including dopaminergic brain areas [47], and dopamine levels increase in both the PVN and SON, the two nuclei where oxytocin is produced, as well as the medial preoptic area, VTA, nucleus accumbens, septum, and the olfactory bulb [47,48]. These areas are all reached by oxytocin neurons. In addition, D2 receptors increase in the nucleus accumbens [49], and stimulation of D1 receptors within the nucleus accumbens or the medial preoptic area has been shown to stimulate the onset of maternal behavior in rats [50]. The increase in oxytocin receptor binding in the medial preoptic area has been suggested to facilitate the dopaminergic neurons within this area to further activate the mesolimbic dopaminergic pathway [51], thereby serving as a rewarding system to facilitate maternal behavior and attachment between mother and offspring [51,52].

In further support of this, an oxytocin antagonist injected into the medial preoptic area, or the VTA, inhibits maternal behavior in rats [53]. In addition, an oxytocin antagonist injected into the nucleus accumbens prolonged the onset of maternal behavior [54].

Correlations between maternal behavior, oxytocin, and dopamine levels have also been reported. Shahrokh et al. showed that rat dams with a high degree of maternal behavior (licking and grooming with the rat pups) had higher oxytocin levels in the preoptic area and the hypothalamus. Dams with a high degree of maternal behavior also had higher dopamine levels in the nucleus accumbens. This difference was abolished when an oxytocin antagonist was administered, indicating that oxytocin mediates the release of dopamine [55].

In a study by Wang et al. in voles, it was demonstrated that a dopamine reuptake inhibitor decreased paternal care as well as the immunoreactivity of oxytocin neurons in the PVN [56].

In humans, a study with fMRI showed an increased activation of oxytocin in the VTA and an increase in oxytocin in cerebrospinal fluid in women when they saw pictures of babies [20,57,58]. In another neuroimaging study by Strathearn, reduced activation of the mesocorticolimbic dopamine pathways was found in mothers with insecure or dismissed attachment [59]. In addition, there are also studies that show correlations between peripheral oxytocin levels and increased activation of the striatum, the nucleus accumbens, and the amygdala. Mothers interacted with or watched pictures of their infants, and the higher the oxytocin levels, the more activation of these brain areas [60,61].

Studies with fMRI have been performed also in fathers, where oxytocin increased the activity in several dopaminergic areas when they were exposed to pictures of their children [62].

### 3.2. Feeding Behavior

Oxytocin and dopamine are released in response to feeding and interact with the anorexogenic/orexogenic neurons within the arcuate nucleus. Both inhibitory and stimulatory effects on food intake and appetite have been reported.

Dopamine is, in particular, released in response to a high-carbohydrate and high-fat diet, and since dopamine is associated with pleasure and reward, it has been speculated that dopamine is involved in overeating [63,64]. The ob/ob mouse, which is characterized by a leptin deficiency, has reduced dopamine levels in the arcuate nucleus [65]. In 1988, Hernandez and Hoebel [64] showed a feeding-induced increase in dopamine in the nucleus accumbens, and it has been suggested that especially the VTA and the mesolimbic dopamine pathway contribute to the ‘rewarding’ aspects of consuming appetizing foods [66,67]. Oxytocin has been demonstrated to modulate the dopaminergic neurons in these areas in experiments on food intake (see, for example, review by Liu et al. [68]).

Oxytocin may both increase and decrease food intake depending on, for example, other hormone levels and whether the individual is in a fed or fasting state [69,70]. In the hypothalamus, both melanocyte-stimulating hormone (MSH) and leptin have been demonstrated to affect oxytocin release from the PVN [71,72]. There seem to be correlations and interactions between GLP-1 and oxytocin, too [73,74]. However, to our knowledge, no changes in the oxytocin levels in the ob/ob mice have been demonstrated [75].

In a human study using positron emission tomography (PET), a correlation between the activity in the striatum and eating chocolate was demonstrated [76].

An altered dopaminergic activity has been demonstrated in studies with PET in both obese and anorectic patients. In obese patients, the availability of D2-receptors was decreased in proportion to their BMI [77,78].

Oxytocin has, as mentioned above, sometimes opposite effects, and in humans, oxytocin administered intranasally has been demonstrated to decrease food intake in obese patients with compulsive eating, whereas it may increase food intake in anorectic patients [79].

In addition, in children with Prader–Willi Syndrome (PWS), a genetic neurodevelopmental disorder where two of the characteristics are obesity and overeating, lower levels of oxytocin have been reported. In some but not all clinical studies in these individuals’ oxytocin administered intranasally decreased food intake [80].

### 3.3. Sexual Behavior

Both oxytocin and dopamine increase in response to sexual arousal and sexual activity. Baskerville and Douglas have suggested an oxytocin dopamine circuit where oxytocin stimulates the mesolimbic dopamine system via oxytocin release in the amygdala, hippocampus, and VTA. This activation, in turn, stimulates incertohypothalamic dopaminergic fibers innervating the medial preoptic area, PVN, and SON and causes further oxytocin release, which in turn activates the “rewarding” mesolimbic dopaminergic pathways [20]. Indeed, oxytocin has been demonstrated to be released in the amygdala and the nucleus accumbens in response to sexual behavior [81].

In male rats, both peripheral, ICV as well as microinjections of oxytocin into the hippocampus, amygdala, VTA, and PVN have been demonstrated to facilitate sexual behavior [38,82,83], and administration of an oxytocin antagonist has been shown to inhibit the penile erection induced by dopamine [84]. In female rats, activation of hypothalamic oxytocin neurons coincides with lordosis behavior [85].

In humans, oxytocin and dopamine increase in response to both sexual arousal and sexual activity [86,87,88], and imaging studies with fMRI have demonstrated an increase in the activity in the VTA in response to oxytocin intranasally and erotic stimuli in both men and women [41,57].

### 3.4. Social Behavior and Pair Bonding

Oxytocin and dopamine are necessary for normal social behavior. They are both released in response to social interaction, and both oxytocin and dopamine knockouts and transgenic mice show abnormal social behavior. Oxytocin and oxytocin receptor knockout mice may not even recognize other mice [89,90,91].

Studies in rats suggest that oxytocin increases social interaction by decreasing the activity within the amygdala [92]. Oxytocin and dopamine receptors are located in the amygdala, which is innervated by both oxytocinergic and dopaminergic neurons, and in particular, neurons from the VTA are of decisive importance here. Hung et al. have demonstrated that stimulation of the oxytocinergic neurons within the PVN or their nerve terminals in the VTA enhanced social behaviors, whereas inhibition of the oxytocinergic nerve terminals in the VTA decreased social behaviors [93]. The medial preoptic area also seems to be involved since an infusion of oxytocin into the medial preoptic area in knockout mice restores several of the social deficits [94]. Behaviors are also influenced by odors. The olfactory bulb is reached by oxytocinergic and dopaminergic neurons, and both oxytocin and dopamine receptors are expressed within the olfactory bulb [20,95,96].

The effects of oxytocin on behavior might become long-lasting since oxytocin administered within 5 min after birth in a mouse model of autism normalized social recognition in a long-term perspective, and in rats, oxytocin administered postnatally induces long-lasting behavioral effects [97,98,99].

Of special interest in these aspects are studies in the monogamous prairie voles. Pair bonding between adult prairie voles has been suggested to involve interactions between oxytocin and dopamine, for example, in the nucleus accumbens and the anterior cingulate cortex. Both these areas are provided with oxytocin receptors, and in a study by Burkett et al., social behavior decreased when an oxytocin receptor antagonist was administered in the region of the anterior cingulate cortex [100]. Additionally, when a dopaminergic antagonistic was administered, the effects of oxytocin on partner preference in female prairie voles were diminished [49]. When the distribution of brain oxytocin receptors in the prairie voles was compared with that of nonmonogamous voles, it was found that the prairie voles had a higher density of oxytocin receptors in, for example, the nucleus accumbens, the prefrontal cortex, and the amygdala [101].

It is worth mentioning that oxytocin levels may also change in response to human-animal interaction. When dogs were separated from their owners, oxytocin levels in the dogs increased when reunited, and an even more sustained effect was observed following tactile contact between the owners and their dogs [102].

There are also several studies in humans where oxytocin is released in response to social cues and increases social salience [103,104]. Moreover, in a study by Grewen et al., support from a partner was linked to higher oxytocin levels in both men and women [105].

In 2005, Kosfeld et al. showed that oxytocin administered intranasally could increase trust in humans [106]. It may also affect decision-making and risk-taking, an effect that in one study was dependent upon outcome predictability [107]. It should be mentioned that these results have not always been possible to replicate. It has even been suggested that intranasally administered oxytocin does not enter the brain but the circulation and that the effects observed by intranasal oxytocin are indirectly mediated by the activation of sensory nerves following the increased circulating levels of oxytocin [108].

However, studies with fMRI in humans have demonstrated increased signaling within the VTA and the nucleus accumbens, areas that are provided with both oxytocin and dopamine receptors, in response to intranasally administered oxytocin (see, for example, Scheele et al. [42]).

In addition, Domes et al. reported that oxytocin administered intranasally to humans decreased activation of the amygdala [109], and Rilling et al. showed that intranasal treatment with oxytocin decreased activity in several brain areas that receive projections from the mesolimbic DA pathways. However, the results differed with gender and whether the performed task was a cooperative task or not [110].

### 3.5. Addiction/Reward

The important role of dopamine and especially of the mesocortical/mesolimbic pathway in drug addiction has been known for a long time (see, for example, Joffe et al. [111]). Oxytocin is also affected by drugs; for example, cocaine decreases oxytocin in the nucleus accumbens but increases oxytocin in the hypothalamus and hippocampus. However, after repeated injections, cocaine reduces oxytocin levels also in the hypothalamus and hippocampus [112]. This change has been suggested to contribute to the impaired social capacity that is often seen in addicts. It is also well known that opioids may inhibit oxytocin release.

Both dopamine and oxytocin are released in response to feeding, sex, and pleasure, and they might be involved in the mechanisms behind the escalation of normal behaviors turning into addiction and abuse. It is well known that L-dopa, which is used in the treatment of Parkinson’s disease, can increase impulsive behaviors such as gambling, eating, and shopping, as described in the review by Friedman in 1994 [113].

Listening to music has been demonstrated to increase activity in several brain regions associated with dopaminergic activity [114]. There are studies showing that oxytocin may also increase [115].

A study published in Nature 1998 found evidence that playing video games released dopamine in the human striatum, and the release was even correlated with the level of performance. The better the performance, the more dopamine activation [116]. In addition, studies have shown lower D2 receptor availability in several dopaminergic areas in individuals with a gaming addiction, and in one study, they even correlated with the number of years of gaming [117]. An effect that can be associated with gaming addiction. In addition, a change in D2 receptors in obese individuals with food addiction has been suggested [77,78].

### 3.6. Anxiety/Obsessive–Compulsive Disorders

Dopamine increases in response to a number of stressors, and the dopaminergic mesolimbic pathway is of central importance for stress-coping behavior (see, for example, Baik [118]). However, in response to chronic stress, dopamine may instead decrease. The effects probably depend on the type of stressor and the studied brain area. For example, Dremencov et al. showed decreased dopamine levels within the hypothalamus in rats exposed to chronic stress, and Chang and Grace used in vivo single neuron recordings in anesthetized rats and reported a decrease in VTA activity in animals subjected to chronic mild stress [119,120].

Similarly, oxytocin may both increase or decrease depending on the type of stress and physiological context. Oxytocin is well known to mediate anti-stress effects (for example, by decreasing cortisol levels and blood pressure). In rats, oxytocin can induce both anxiolytic-like and sedative effects depending on dose and route of administration but also depending on the stage of the estrous cycle [121,122]. The effects of oxytocin administered postnatally to pigs and rats may even induce lifelong behavioral changes [123,124,125].

Oxytocin knockout mice express an increased level of anxiety [90,91]. In contrast, in a mouse model lacking a dopamine transporter in the nucleus accumbens, anxiety and depression-like behavior decreased [126].

A nice study showing oxytocin release in response to social interaction giving rise to the anti-stress effects of oxytocin was made by Cardoso et al. In an experimental setting, children exposed to a situation of social stress were tested. As a result, children who could see or hear their mothers’ voices had higher levels of oxytocin and lower levels of cortisol compared to children who had no contact with their mothers [127].

Oxytocin administered intranasally has been demonstrated to decrease anxiety in humans (see, for example, Guastella et al. [128]).

Dopamine is mainly connected to alertness, and sometimes oxytocin and dopamine may have totally opposite effects. Low oxytocin levels have been suggested to contribute to the symptoms of OCD, and differences in the oxytocin receptor gene in individuals with OCD have been found [129]. In contrast, an increase in dopamine levels has been linked to the worsening of the symptoms of OCD [130].

### 3.7. Depression and Bipolar Disorders

Low levels of oxytocin and dopamine have been associated with depression. In animal models of depression, dopamine levels within the VTA are changed (reviewed by Kaufling 2019 [131]), and oxytocin has been found to induce similar effects as antidepressant medication [132]. Interestingly, oxytocin is released in response to selective serotonin reuptake inhibitors (SSRI) [133,134].

Deviations in the regulation of dopamine transmission in the prefrontal cortex have been reported in depression and bipolar disorders [135]. In a meta-analysis of studies of dopamine in patients with depression from Mizuno et al., 2023, the results were not totally consistent, but they indicated lower availability of DAT and D1-receptors in the striatum [136]. There are also studies that show an increased binding to D2 receptors in the same area [137] Indeed, antidepressant effects in response to dopamine agonists have been demonstrated in humans [138], and Bupropion, a combined dopamine and noradrenaline reuptake inhibitor, is in clinical use.

In addition, oxytocin levels may be lower in female patients with depression [139], and in some but not all studies, intranasal administration of oxytocin has also been reported to improve the symptoms of depression [140,141,142]. In a recent study by Baron-Cohen et al. [143], oxytocin intranasally increased mood in postpartum women with low mood (postpartum blues) but not in mothers with postpartum depression.

### 3.8. Autism Spectrum Disorders

Oxytocin has been suggested to play a role in the symptoms related to autism spectrum disorders in humans (see, for example, a review by Yoon and Kim [144]). Correlations between specific oxytocin receptor polymorphisms and autism have been suggested [145,146]. There are several mouse models of ASD. For example, in a mouse model named Shank3BKO with decreased oxytocin immunoreactivity within the PVN, oxytocin increases social behavior and dopaminergic activity within the VTA. In addition, when oxytocin receptors in dopaminergic neurons were blocked, no change in social behavior was seen [147].

In another mouse model of autism, oxytocin administered neonatally restored social behavior [148].

In a meta-analysis by Moerkerke et al. [149], lower oxytocin levels were found in children but not in adolescents with autism, and in some studies, the administration of oxytocin has been reported to improve social behavior in these individuals.

Several studies in humans have been performed, and for example, in one study, oxytocin infusion reduced repetitive behavior in individuals with ASD [150], and in another study, oxytocin was found to reduce an increased activity in the amygdala in response to angry faces in the participants with ASD [151]. There are links between oxytocin and dopamine also in ASD. In a study where improved learning in young adults with ASD was seen in response to oxytocin, there was also an increase in the signaling of nucleus accumbens [152]. Thus, dopamine might also be involved in the pathophysiology of ASD, and a change in the dopamine transporter system has been suggested [153].

Usually, no change in peripheral dopamine levels is found, but there are studies reporting both increases and decreases [154,155]. Important to remember here is that dopamine is also produced outside the brain, and it does not cross the blood–brain barrier. However, individuals with ASD are a heterogenic group, and ADHD and OCD are not uncommon in these individuals. In a neuroimaging study by PET, there was a 39% reduction in the uptake of radiolabeled dopa in the anterior medial prefrontal cortex in ASD compared to controls [156]. In a review by Di Carlo and Wallace, several imaging studies are presented, and most of them show changes in dopaminergic areas in individuals with ASD [157].

### 3.9. Attention Deficit Hyperactivity Disorder (ADHD)

Lower levels of oxytocin and dopamine have been associated with ADHD, which is characterized by less attention, hyperactivity, and impulsivity. Several studies demonstrate lower levels of dopamine within the CNS, and this effect has been suggested to be due to changed levels in dopamine transporter proteins and/or dopamine receptors.

Dopamine transporter knockout mice (DAT-KO) and spontaneously hypertensive rats (SHR) have been used as animal models of ADHD. DAT-KO mice are hyperactive and have attention deficits. These rats have changes in their dopamine receptors, and their symptoms are alleviated by methylphenidate. The SHR is a rat that was originally used for studies of high blood pressure, but it also shows attention deficits, hyperactivity, and impulsiveness. In this rat, decreased dopamine levels in, for example, the nucleus accumbens, as well as a change in dopamine receptors, have been demonstrated [158,159,160]. Interestingly, the SHRs have lower oxytocin levels [161], and there are studies suggesting that an increase in dopamine in response to psychostimulants, such as methylphenidate, is mediated through oxytocin receptors. In a study by Hersey et al., rats pretreated with oxytocin showed a significantly larger increase in dopamine in response to methylphenidate, and this effect was abolished if an oxytocin antagonist was administered [162].

The effects of psychostimulants, such as methylphenidate, used in the treatment of ADHD, are mainly thought to be induced through an increase in dopamine levels, although noradrenaline levels are also affected. However, in a study with brain imaging (combined PET and MR) from del Campo et al., there were no significant differences in dopamine levels in response to methylphenidate between individuals with ADHD and healthy controls. Instead, individuals with ADHD showed reduced gray matter volume in the fronto–striato–cerebellar and limbic regions [163].

Another mechanism that has been suggested, besides a change in dopamine transporter proteins, is changes in dopamine receptor availability, and in a systematic meta-analysis of dopamine receptor genes by Wu et al. [164], associations between specific variants of the D4 receptor and ADHD were found.

In addition, there are studies showing lower levels of oxytocin in children with ADHD [165,166]. In a study by Shachar et al., 2020, this difference in oxytocin levels disappeared and became similar to those in healthy controls after the administration of methylphenidate [165]. However, a study in adults did not show any correlation between ADHD-related symptoms and oxytocin levels [167].

Last, worth mentioning is that there are a few studies suggesting that children prenatally exposed to high levels of obstetric oxytocin could have an increased risk of developing ADHD, suggestions that could not be confirmed in a large study by Stokholm et al. [168]. Instead, in a very recent study by Jallow et al., breastfeeding—a period characterized by high oxytocin levels—decreased the risk of ADHD [169]. Since oxytocin penetrates the placenta poorly, any negative effects on the fetus should be indirectly mediated by a deranged pattern of urine contractions [170].

### 3.10. Schizophrenia

It is well known that dopamine is involved in the pathophysiology of schizophrenia, and classical antipsychotic treatments are assigned to affect dopamine receptors. Especially the mesocortical and mesolimbic pathways seem to be important, and it is thought that these pathways are dysfunctional in schizophrenia [171]. However, functional changes in the dopaminergic brain areas have been difficult to show in neuroimaging studies (see, for example, a review by Schulz et al. [172]). In recent years, other neurotransmitters have also been found to be involved in the symptoms of schizophrenia. Changes in serotonin, gamma-aminobutyric acid (GABA), glutamate, acetylcholine, and oxytocin levels have been demonstrated. [173,174]. There are animal models that have been used to study symptoms of schizophrenia, but since it is a complex disorder, it is difficult to draw any conclusions regarding the possible role of oxytocin together with dopamine from them (see, for example, a review by Jones et al. [175]). However, they have shown that oxytocin may modulate the activity in the mesolimbic dopaminergic pathway, as already discussed above, but in the case of schizophrenia, oxytocin may decrease the activity, supported by the fact that oxytocin has the capacity to inhibit the hyperactivity in the nucleus accumbens induced by drugs such as cocaine, methamphetamine, and phencyclidine [175].

In humans, oxytocin has been suggested to particularly have a role in the negative symptoms of schizophrenia, i.e., social withdrawal and reduced emotional expression. Higher plasma oxytocin levels in schizophrenic patients have been found to be associated with lower severity of symptoms [176], and in a study by Bradley et al., oxytocin administered intranasally increased eye contact in individuals with schizophrenia [177]. Clozapine, a second-generation antipsychotic drug, but not haloperidol, increases oxytocin [36]. However, in another study of humans with schizophrenia, lower oxytocin levels were correlated both to the antipsychotic drugs of the second generation and to more negative symptoms [178]. Studies with oxytocin administered intranasally have also been performed in patients with schizophrenia, with improvements in symptoms in some but not all of the studies [179].

## 4. Conclusions

In conclusion, oxytocin and dopamine have several effects in common and modulate each other´s effects. The effects may deviate depending on the situation and physiological context. As discussed above, the effects might also differ, for example, between sexes, different ages, and between lean and obese individuals. In a recent review of neuroimaging studies and oxytocin administered intranasally, Prosyshyn et al. concluded that the effects often were opposite in females versus males [180]. When oxytocin has been administered intranasally to individuals with PWS, the effects seem to be most pronounced in the youngest [80], and different effects of oxytocin administered intranasally may also be induced depending on weight and eating behavior (see, for example, Iovino et al. [79]). We have not discussed personality disorders in the present paper since the role of oxytocin and dopamine seems to be more unclear. For example, it could be expected that individuals with antisocial personality disorder would have lower oxytocin levels and/or could be treated with oxytocin. However, in a review by Gedeon et al., oxytocin did not have any convincing effect, and in some studies, even effects that were opposite to what was expected were found [181].

Generally, oxytocin increases dopaminergic activity (Table 2), but it seems that oxytocin has the capacity to decrease dopaminergic activity as well. For example, as discussed above, the results may differ depending on gender, as demonstrated by Rilling et al., who showed that intranasal treatment with oxytocin sometimes might induce opposite effects in men and women [110]. In animal models of schizophrenia, oxytocin seems to decrease dopaminergic activity, for example, in the nucleus accumbens, an effect that has also been demonstrated in response to drugs such as cocaine, methamphetamine, and phencyclidine [175].

Dopamine release and turnover are regulated by different dopamine receptors, whereas regarding oxytocin, only one oxytocin receptor has been demonstrated. Mechanisms behind the opposite effects of oxytocin on dopaminergic activity might be explained by the expression of the receptors, where dopamine, as mentioned, has different receptor subtypes and interactions through receptor complexes between oxytocin and dopamine-receptors [39,40]. It is also possible that different effects are induced depending on variability in the oxytocin gene, as suggested by Love et al. [182], or by different oxytocin fragments. The oxytocin molecule is degraded into different smaller fragments, and some of them are active.

The c-terminal tripeptide of oxytocin (prolyl-leucyl-glycinamide (PLG)), also known under the name MIF-1, shares some but not all of the effects of oxytocin, and it has been demonstrated to affect D2 receptors [183]. PLG decreases locomotor activity in rats, and in high doses, it decreases oxytocin levels in plasma [184]. In addition, another fragment, oxytocin 4–9, administered in a mouse model of autism, increased social behavior [185].

In addition, the many and sometimes almost opposite effects might also be explained by the interactions of oxytocin and dopamine with other hormones, neurotransmitters, and receptors. For example, serotonin (5-HT), opioids, GABA, the hypothalamic–pituitary–adrenal (HPA) axis, cholecystokinin (CCK), and alpha2-adrenoreceptors have all been shown to interact with oxytocin and dopamine.

Oxytocin in high doses might also interact with vasopressin receptors, thereby sometimes causing opposite effects (see, for example, a review by Rae et al. [186]).

Worth mentioning is also that both SSRI and 5HT1-agonists increase oxytocin levels [133,134].

Oxytocin and dopamine are both necessary for behavior, and they both have important roles in many psychiatric and behavioral disorders. In general, oxytocin and dopamine often have similar effects, but in diagnoses characterized by repetitive and compulsive behavior, such as OCD and schizophrenia, oxytocin and dopamine levels usually seem to be the opposite.

## 5. Future Directions

Together with the many different and sometimes opposite effects, it is difficult to use dopamine and oxytocin as treatments. It is also important that the effects of both oxytocin and dopamine might be long-lasting. The early separation between mother and offspring may induce lifelong changes in both oxytocin and dopamine levels [187], and in animal studies, postnatally administered oxytocin induces long-lasting, perhaps lifelong effects. When administered postnatally, oxytocin has been found to induce long-lasting effects on growth, behavior, and blood pressure, as well as the HPA-axis and alpha 2-adrenoreceptors [98,99,123,124,125].

Given the many effects and interactions of oxytocin and dopamine, particularly in the context of the integrated and complex effects of oxytocin and dopamine acting together, more studies during different situations and conditions are needed. There are already many dopamine analogs, reuptake inhibitors, and dopamine-releasing agents available and also in clinical use. The challenge is to find more specific agents acting in specific parts of the brain.

When it comes to oxytocin, trials with oxytocin and the more long-acting analog carbetocin in patients with autism and patients with PWS are ongoing [188,189,190,191,192]. New nonpeptide oxytocin receptor agonists are upcoming (see, for example, Frantz et al. [189]), and some of them have shown promising effects in animal models of social disorders. Recently, melanocortin receptor agonists were demonstrated to have specific effects on oxytocin release from the PVN and during certain circumstances, especially on oxytocin release in the nucleus accumbens [190]. Thus, hopefully, new and more specific oxytocin and dopamine agonists or analogs will be available in the future.

## Figures and Tables

**Table 1 biomedicines-12-02440-t001:** Characteristics of oxytocin and dopamine.

	Oxytocin	Dopamine
**Chemical structure**	Nonapeptide	Monoamine
**Receptors**	One main receptor	Two main types divided into five subtypes (D1–D5)
**Main production sites within the brain**	Hypothalamus	Substantia nigra, ventral tegmental area
**Main behavioral effects**	Maternal, sexual, and social behavior	Rewarding and sexual behavior
**Typical emotional effects**	Anxiolytic, calming, and relaxing	Pleasure, motivation, and alertness

**Table 2 biomedicines-12-02440-t002:** Behaviors and disorders modulated by oxytocin and dopamine.

Parental behavior	OXT ↑ → DA ↑
Sexual behavior	OXT ↑ → DA ↑
Social behavior/pair bonding	OXT ↑ → DA ↑
Feeding behavior	OXT ↑ → DA ↑
Addiction/Reward	OXT ↓↑ ↔ DA ↑
Anxiety	OXT ↓ ↔ DA ↑↓
Depression	OXT ↓ ↔ DA ↓
ASD	OXT ↓ → DA ↓
ADHD	OXT ↓ ↔ DA ↓
Schizophrenia	OXT ↓ ↔ DA ↑↓

The right column shows if oxytocin (OXT) primarily stimulates dopamine (DA) or vice versa and if their levels generally increase or decrease.

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
