# Peer review of "Interactions of Oxytocin and Dopamine—Effects on Behavior in Health and Disease"

_biomedicines, 2024, doi:10.3390/biomedicines12112440_

Round 1

Reviewer 1 Report

Comments and Suggestions for Authors

1. Please check the overall abbreviation in the manuscript.

2. All the references should be categorized in table form with different needs, it is more convincing for readers.

3. The Conclusions and Future Directions should be separated, it is too long to read.

4. How about the serotonin and endorphins? Why focus on the oxytocin and dopamine? I think the review will be more important if include these hormones.

1. Comprehensive Coverage of Literature
Comment: The review article presents a detailed examination of the interactions between oxytocin and dopamine, particularly in the context of behavior and mental health disorders. However, it would benefit from incorporating more recent studies, especially those published within the last 3-5 years. This would ensure that the review provides the most up-to-date insights, particularly regarding new therapeutic developments like non-peptide oxytocin receptor agonists and recent clinical trials on oxytocin-based treatments.

2. Depth of Mechanistic Insights
Comment: The article introduces the interactions between oxytocin and dopamine well, but the mechanistic explanations could be expanded. Specifically, the discussion on how oxytocin influences dopamine receptor signaling across different brain regions (e.g., VTA, nucleus accumbens) could be more in-depth. A more thorough exploration of the signaling pathways and receptor-level interactions would greatly enhance the scientific value of the review.

3. Clinical Implications and Future Directions
Comment: While the review touches on the clinical relevance of oxytocin-dopamine interactions, this section feels underdeveloped. It would be helpful to include a more detailed discussion of ongoing clinical trials and the challenges of translating these findings into therapies. Additionally, outlining potential future research directions—such as the development of more specific oxytocin receptor agonists—would provide a clearer roadmap for readers.

4. Organization and Flow
Comment: The flow of the article can be improved by ensuring smoother transitions between sections. Some areas, such as the behavioral effects of oxytocin and dopamine, feel somewhat disjointed from earlier mechanistic discussions. Strengthening these connections would make the article more cohesive and easier to follow.

5. Terminology and Clarity
Comment: There are occasional inconsistencies in the use of terms, particularly in referring to receptor subtypes and signaling pathways. Ensuring that these are defined and used consistently throughout the article will help improve clarity and reduce potential confusion among readers.

6. Reference Updates
Comment: Some of the references are slightly outdated. While older studies provide important context, replacing or supplementing these with more recent research findings will keep the review current and enhance its relevance to the field.

7. Formatting and Typographical Errors
Comment: The article contains a few typographical and formatting errors, which should be addressed in the next revision. Correcting these will improve the overall presentation and readability of the article.

Comments on the Quality of English Language

Fine

Author Response

Ref 1

Thank you for your valuable comments. Please find all answers below, point by point. Changes in manuscript are highlighted in yellow (except minor changes for example of typos.

  1. Please check the overall abbreviation in the manuscript.

Done

2. All the references should be categorized in table form with different needs, it is more convincing for readers.

Sorry but we are not sure what you mean here? This review is not supposed to cover this area in total. If we should categorize all the references based on type of behavior, specie or result? with needs it would be a very large table. In this review, instead of trying to cover all new studies which would be impossible (if not choosing just one of the behaviors) we have selected papers representing old fundamental or milestone papers and papers representing different aspects. However as you suggested we have included some more recent studies and planned studies according to NIH clinicaltrials.gov (they are highlighted in yellow in the manuscript).
3. The Conclusions and Future Directions should be separated, it is too long to read.
They are now separated
4. How about the serotonin and endorphins? Why focus on the oxytocin and dopamine? I think the review will be more important if include these hormones.
The aim of this paper was to focus on oxytocin and dopamine and the title of the special issue is Neuropeptides, Dopamine and Their Interactions in Neuroscience. But we agree it would be a very interesting paper (and even more complex).

1. Comprehensive Coverage of Literature
Comment: The review article presents a detailed examination of the interactions between oxytocin and dopamine, particularly in the context of behavior and mental health disorders. However, it would benefit from incorporating more recent studies, especially those published within the last 3-5 years. This would ensure that the review provides the most up-to-date insights, particularly regarding new therapeutic developments like non-peptide oxytocin receptor agonists and recent clinical trials on oxytocin-based treatments.

We have included some more recent studies and also recent (ongoing) clinical trials.

2. Depth of Mechanistic Insights
Comment: The article introduces the interactions between oxytocin and dopamine well, but the mechanistic explanations could be expanded. Specifically, the discussion on how oxytocin influences dopamine receptor signaling across different brain regions (e.g., VTA, nucleus accumbens) could be more in-depth. A more thorough exploration of the signaling pathways and receptor-level interactions would greatly enhance the scientific value of the review.

We agree this is very interesting but a deeper discussion is beyond the focus of this review. However, we added a recent review about the receptor-receptor interactions which we agree need more focus.

  1. Clinical Implications and Future Directions
    Comment: While the review touches on the clinical relevance of oxytocin-dopamine interactions, this section feels underdeveloped. It would be helpful to include a more detailed discussion of ongoing clinical trials and the challenges of translating these findings into therapies. Additionally, outlining potential future research directions—such as the development of more specific oxytocin receptor agonists—would provide a clearer roadmap for readers.

We agree this would be interesting but as you say it is a difficult topic. We have rewritten this part of the manuscript. As far as we know no non-peptidergic oxytocin analogue is in clinical trial.
4. Organization and Flow
Comment: The flow of the article can be improved by ensuring smoother transitions between sections. Some areas, such as the behavioral effects of oxytocin and dopamine, feel somewhat disjointed from earlier mechanistic discussions. Strengthening these connections would make the article more cohesive and easier to follow.

Hope the transitions are smoother now.

5. Terminology and Clarity
Comment: There are occasional inconsistencies in the use of terms, particularly in referring to receptor subtypes and signaling pathways. Ensuring that these are defined and used consistently throughout the article will help improve clarity and reduce potential confusion among readers.

We hope this is more clear now
.
6. Reference Updates
Comment: Some of the references are slightly outdated. While older studies provide important context, replacing or supplementing these with more recent research findings will keep the review current and enhance its relevance to the field.

We prefer to keep the old and fundamental studies which also as mentioned by one of the other referee is a strength of this manuscript. But as suggested we have added some very recent studies including clinical trials.
7. Formatting and Typographical Errors
Comment: The article contains a few typographical and formatting errors, which should be addressed in the next revision. Correcting these will improve the overall presentation and readability of the article.

Hope this is better now.

Reviewer 2 Report

Comments and Suggestions for Authors

In the Abstract, lines 15-17 state, 'Although, being structurally totally different, oxytocin a peptide and dopamine a monoamine, they have a number of similar effects. They affect each other's release and receptors and are synthesized and have G-protein-coupled receptors both in the brain and in the periphery.' It mentions the significant structural differences between oxytocin and dopamine, but this is not further explained in the main text. Does 'G-protein-coupled receptors' refer to how oxytocin and dopamine, as hormones or neurotransmitters, regulate through these receptors? Please elaborate on this in the main text.

In Section 1. 'The Discovery of Oxytocin and Dopamine,' the description of adrenalin, vasopressin, and oxytocin in lines 26-34 is unclear and lacks coherence. What is the author trying to convey? In lines 38-42, the explanation of oxytocin as both a hormone and a neurotransmitter appears after the introduction of dopamine in lines 35-37, only to then return to oxytocin. Then, in lines 43-47, the structure of dopamine is followed by a discussion of the Nobel Prize related to oxytocin, before shifting back to dopamine’s Nobel Prize in lines 48-51. Please provide a detailed introduction to oxytocin as both a hormone and neurotransmitter first, then describe dopamine, and finally compare their similarities and differences. This would make it easier for readers to follow, rather than alternating between the two.

Lines 73-76 “Oxytocin neurons project to many areas with dopaminergic neurons or areas reached by dopaminergic neurons...” Please cite reference?

Lines 83-91 “Dopamine has several distinct pathways within the brain: the nigrostriatal pathway, from the substantia nigra to striatum...” Please cite reference?

Lines 210-211 “Oxytocin and dopamine are released in response to feeding and interact with the...” Please cite reference?

Could the similar descriptions in lines 100-101, 114-115, 279, 378… be made consistent?

In Section 3.2 'Feeding behavior,' could the descriptions of dopamine and oxytocin be clearly separated? Should lines 213-219 and 226-234 be combined for clarity? Additionally, in the other paragraph about oxytocin, is it in the third paragraph where oxytocin might influence dopamine, or do they mutually affect each other?

In Section 3. 'Behavioral Effects of Oxytocin and Dopamine,' there are many individually cited references, such as in lines 193-195, 230-231, 296-298, and 406-407. Could these be reorganized into Table 1?

Finally, could a table or figure be provided to summarize the structural differences between oxytocin and dopamine, as well as their functions or distinctions as hormones? Additionally, please show their respective expression in different brain regions as neurotransmitters, and how they influence each other.

Comments on the Quality of English Language

Minor editing of English language required

Author Response

Thank you for your valuable comments. Please find the answers below, point by point.

Changes in the manuscript are highlighted in yellow (except of minor changes for example of typos)
In the Abstract, lines 15-17 state, 'Although, being structurally totally different, oxytocin a peptide and dopamine a monoamine, they have a number of similar effects. They affect each other's release and receptors and are synthesized and have G-protein-coupled receptors both in the brain and in the periphery.' It mentions the significant structural differences between oxytocin and dopamine, but this is not further explained in the main text. Does 'G-protein-coupled receptors' refer to how oxytocin and dopamine, as hormones or neurotransmitters, regulate through these receptors? Please elaborate on this in the main text.

This part of the abstract Is now rewritten. We omitted the G-protein-coupled receptors here since it is not important.

In Section 1. 'The Discovery of Oxytocin and Dopamine,' the description of adrenalin, vasopressin, and oxytocin in lines 26-34 is unclear and lacks coherence. What is the author trying to convey? In lines 38-42, the explanation of oxytocin as both a hormone and a neurotransmitter appears after the introduction of dopamine in lines 35-37, only to then return to oxytocin. Then, in lines 43-47, the structure of dopamine is followed by a discussion of the Nobel Prize related to oxytocin, before shifting back to dopamine’s Nobel Prize in lines 48-51. Please provide a detailed introduction to oxytocin as both a hormone and neurotransmitter first, then describe dopamine, and finally compare their similarities and differences. This would make it easier for readers to follow, rather than alternating between the two.

Thank you for pointing this out. This is meant to be a historical introduction to oxytocin and dopamine. We have changed the subtitle and we also made some changes in the text to make it more clear and chronological.

Lines 73-76 “Oxytocin neurons project to many areas with dopaminergic neurons or areas reached by dopaminergic neurons...” Please cite reference?

Included in the end of the paragraph

Lines 83-91 “Dopamine has several distinct pathways within the brain: the nigrostriatal pathway, from the substantia nigra to striatum...” Please cite reference?
included
Lines 210-211 “Oxytocin and dopamine are released in response to feeding and interact with the...” Please cite reference?
This is a short introduction and the references are included in the following text.

Could the similar descriptions in lines 100-101, 114-115, 279, 378… be made consistent?
Line 100-101 is examples of where oxytocin increases dopamine and will be discussed further in the manuscript

Line 114-115 – there are so many papers about what triggers the release of oxytocin and dopamine and therefore we prefer to have reviews

Line 279 We added two extra references here and kept one of the already included reviews.

Line 378
Corrected. The reference is a review of different animal models.

In Section 3.2 'Feeding behavior,' could the descriptions of dopamine and oxytocin be clearly separated? Should lines 213-219 and 226-234 be combined for clarity?
Additionally, in the other paragraph about oxytocin, is it in the third paragraph where oxytocin might influence dopamine, or do they mutually affect each other?
This part is now rewritten.

In Section 3. 'Behavioral Effects of Oxytocin and Dopamine,' there are many individually cited references, such as in lines 193-195, 230-231, 296-298, and 406-407. Could these be reorganized into Table 1?

We prefer to not have references in the table because it would be difficult to overview with so many references pointing sometimes in the same direction, sometimes opposite depending on gender, age, species, context etc and sometimes are these different and opposite effects discussed in the same reference.

Finally, could a table or figure be provided to summarize the structural differences between oxytocin and dopamine, as well as their functions or distinctions as hormones? Additionally, please show their respective expression in different brain regions as neurotransmitters, and how they influence each other.

A new table is included. Since oxytocin and dopamine are interacting in many brain areas and the interactions not always are the same (depending on gender, situation, species, experimental or clinical situation, healthy or non-healthy etc) we prefer not to try to include any picture of all the possible interactions.

Reviewer 3 Report

Comments and Suggestions for Authors

The Review article by Petersson and Uvnäs-Moberg describes the role of oxytocin and dopamine in pathologic and physiologic conditions, as well as their interactions. The paper is well-written, organized in a systematic manner and updated with recent literature. Some minor Issues should be dealt with, before publication.

- A graphical abstract should be provided to visually summarize the review content to the readers.

- All the acronyms must be defined at first use in the text (i.e. "SSRI" at line 394, "OXT" at line 403, HPA at line 540)

The Review article by Petersson and Uvnäs-Moberg describes the role and interactions of dopamine and oxytocin in physiologic and pathologic conditions. The paper is relevant in the field because it provides a concise review of the updated scientific literature. In contrast with other Review articles dealing with the same topic, this paper offers an overview on the major functions regulated by dopamine and oxytocin, citing more specific literature for further in-depth analyses. References are well-assorted in terms of updated literature and milestone-papers.

The methodology exploited herein is that of narrative reviews, not systematic ones, but in this reviewer opinion, it has been used effectively to describe the main knowledge in this research field and future needs still to be addressed. As already stated, a graphical abstract or a summarizing figure should be provided, since the visual part of the paper is lacking. Comments on the Quality of English Language

Please carefully revise English Language throughout the manuscript, avoiding typos (e.g. lines 40, 168, 295, 466, 531). Check punctuation as well.

Author Response

The Review article by Petersson and Uvnäs-Moberg describes the role of oxytocin and dopamine in pathologic and physiologic conditions, as well as their interactions. The paper is well-written, organized in a systematic manner and updated with recent literature. Some minor Issues should be dealt with, before publication.
Thank you very much for your valuable comments which we appreciate a lot. Please see the answers below point by point. Changes in the manuscript are highlighted in yellow (except minor changes such as typos)
- A graphical abstract should be provided to visually summarize the review content to the readers.
Graphical abstracts are nice but I do not think this paper uses graphical abstracts?
- All the acronyms must be defined at first use in the text (i.e. "SSRI" at line 394, "OXT" at line 403, HPA at line 540)

Done

The Review article by Petersson and Uvnäs-Moberg describes the role and interactions of dopamine and oxytocin in physiologic and pathologic conditions. The paper is relevant in the field because it provides a concise review of the updated scientific literature. In contrast with other Review articles dealing with the same topic, this paper offers an overview on the major functions regulated by dopamine and oxytocin, citing more specific literature for further in-depth analyses. References are well-assorted in terms of updated literature and milestone-papers.

The methodology exploited herein is that of narrative reviews, not systematic ones, but in this reviewer opinion, it has been used effectively to describe the main knowledge in this research field and future needs still to be addressed. As already stated, a graphical abstract or a summarizing figure should be provided, since the visual part of the paper is lacking.

Thank you very much, we appreciate this comment a lot.

A new table is included to illustrate the main characteristics of oxytocin and dopamine.

Comments on the Quality of English Language
Please carefully revise English Language throughout the manuscript, avoiding typos (e.g. lines 40, 168, 295, 466, 531). Check punctuation as well.

Thank you for your thoroughness. Hope we found all.

Round 2

Reviewer 1 Report

Comments and Suggestions for Authors

The concerns have been well addressed.

Comments on the Quality of English Language

Fine.